# Smoking as a risk factor for lower extremity peripheral artery disease in women compared to men: A systematic review and meta-analysis

Ying Xu[1,2]*, Anna Louise Pouncey[3], Zien Zhou[1], Mark Woodward[1,4], Katie Harris[1]

1 Faculty of Medicine, The George Institute for Global Health, University of New South Wales, Sydney, New South Wales, Australia, 2 Faculty of Medicine, Australian Institute of Health Innovation, Centre for Health Systems and Safety Research, Health and Human Sciences, Macquarie University, Sydney, New South Wales, Australia, 3 Faculty of Medicine, Department of Vascular Surgery, Division of Surgery and Cancer, Imperial College London, QEQM, St Mary's Hospital, London, United Kingdom, 4 The George Institute for Global Health, School of Public Health, Imperial College London, London, United Kingdom

* yxu1@georgeinstitute.org.au

**Data Availability Statement:** All extracted data including those used for analyses are presented in the manuscript or supplementary.

## Abstract

### Background

To investigate whether the relationship between smoking and peripheral artery disease (PAD) differs by sex (PROSPERO CRD42022352318).

### Methods

PubMed, EMBASE, and CINAHL were searched (3 March 2024) for studies reporting associations between smoking and PAD in both sexes, at least adjusted for age. Data were pooled using random effects. Between-study heterogeneity was examined using $I^2$ statistic and Cochran's Q test. Newcastle-Ottowa Scale was adopted for quality assessment.

### Results

Four cohort studies (n = 2,117,860, 54.4% women) and thirteen cross-sectional studies (n = 230,436, 59.9% women) were included. In cohort studies, former and current smokers had higher risk of PAD than never smokers. Compared to those who never or previously smoked, women current smokers (relative risk (RR) 5.30 (95% confidence interval 3.17, 8.87)) had higher excess risk of PAD than men (RR 3.30 (2.46, 4.42)), women-to-men ratio of RR 1.45 (1.30, 1.62)($I^2$ = 0%, $p$ = 0.328). In cross-sectional studies, risk of PAD was higher among former and current compared to never smokers, more so in men, women-to-men ratios of odds ratio: 0.64 (0.46, 0.90)($I^2$ = 30%, $p$ = 0.192), 0.63 (0.50, 0.79)($I^2$ = 0%, $p$ = 0.594), respectively. For both sexes, risk of PAD was higher among current smokers compared to those who were not currently smoking. Cohort studies and five cross-sectional studies were of good quality, scoring 6 to 8 of a possible maximum 9 points. Eight cross-sectional studies scored 2 to 5.

**Funding:** This work is supported by the GeorgeThink, an institutional seed funding award, from The George Institute for Global Health. ALP is supported by a National Institute for Health and Care Research (NIHR) Doctoral Fellowship (NIHR301767). ZZ is supported by the NSW Cardiovascular Elite Postdoctoral Researcher Grant (H23/15813) from NSW Health, Australia. MW is supported by an Australian NHMRC Investigator Grant, Leadership 2 (APP1174120), and Program Grant (APP1149987). The funders had no role in study design; collection, analysis, and interpretation of data; writing the report; and the decision to submit the report for publication.

**Competing interests:** MW has done recent consultancy for Amgen and Freeline outside the submitted work, no support from any organization, for the submitted work, no other relationships or activities that could appear to have influenced the submitted work. All other authors have nothing to declare.

## Discussions

Further research is required to elucidate sex differences in the relationships between smoking and PAD, as the current evidence is limited and mixed. Tobacco-control programs should consider both sexes.

## Introduction

The importance of cardiovascular risk in women is under-recognized, especially for peripheral artery disease (PAD). Due to higher rates of asymptomatic disease or atypical symptoms in women with PAD, they are often diagnosed at later stages of the disease, and thus are less likely to receive interventions to prevent advanced PAD and adverse outcomes of amputation, coronary heart disease (CHD), and stroke [1]. Smoking is an important risk factor for PAD [2]. The increased risk among smokers for PAD [2] is similar to that for CHD [3]. For people with PAD who smoke, smoking cessation is recommended as a first-line treatment with highest level of evidence [4]. Nonetheless, tobacco control is a challenge and, in many countries, decreases in the prevalence of smoking has slowed [5]. Women are disadvantaged, as while the prevalence of smoking among men decreased significantly in 135 countries (66% of 204 countries and territories) between 1990 and 2019, a significant decrease in women was seen in only 68 countries (33%) [5]. Further, women smokers have a 25% greater excess risk of CHD compared to men smokers [3], possibly due to the higher amount of toxic agents women get from same number of cigarettes as men [3,6]. However, it is unclear whether women smokers also have greater excess risk of PAD than their men counterparts, or who (women or men) benefit more from smoking cessation. Although magnitudes of associations between smoking and PAD did not differ between studies that recruited only women, only men, or both [2], it does not necessarily mean there was no sex differentiated risk of PAD among women versus men smokers. Thus, we conducted a systematic review with meta-analyses to: 1) investigate whether the increased risk of PAD related to smoking is different in women and men, accounting for other key risk factors and adopting within-study comparisons; 2) determine how smoking intensity and/or duration (e.g., years and daily amount of smoking, pack-years, years since quit among former smokers) and measures of toxic chemicals impact the associations; and 3) whether quitting confers same benefit in women as in men.

## Methods

The protocol of this review was registered in PROSPERO (CRD42022352318). The review is reported according to the Preferred Reporting Items for Systematic Reviews and Meta-Analyses (PRISMA) checklist and Meta-analysis Of Observational Studies in Epidemiology (MOOSE) guideline (S1A-S1C File).

### Inclusion and exclusion criteria

Peer-reviewed cohort, case-cohort, and cross-sectional studies that reported associations between smoking status, intensity and duration, or other measures of smoking, and PAD in women and men, and had at least adjusted for age, were included. Population-based, community, or clinical samples of people at any age were eligible. Studies that recruited only women or men were excluded.

Active cigarette, pipe, and cigar smoking was investigated in the current work, but not passive (cigarette, pipe, cigar) smoking, nor active or passive smoking of e-cigarettes/vapes. Smoking status was categorized and compared as any of the following: 1) (current versus never) and/or (former versus never); 2) current smokers versus non-smokers (former or never); and 3) current versus former. PAD was defined using diagnostic or procedure codes, cut-offs of ankle-brachial index (ABI, e.g., < or ≤0.9), standard questionnaires on intermittent claudication, or angiography.

Outcomes were required to be reported as any of hazard ratio (HR), relative risk (RR), odds ratio (OR), with estimates of the variance, e.g., 95% confidence intervals (CIs) or standard errors. Studies adopting only a backward stepwise selection procedure to define two sets of risk factors separately in women and men were excluded, unless same variables relevant to smoking were retained in both women and men.

## Search strategy and screening

PubMed, EMBASE (Ovid), and CINAHL (EBSCOhost) were searched, by one reviewer (YX), from inception to 3 March 2024 (S1 Table). When possible (depending on the database), the search was restricted to journal articles and "human". No language restrictions were applied. Search terms relevant to smoking, PAD, and sex were used as free text or controlled vocabulary (e.g., medical subject headings (MeSH), EMTREE) in each database.

One reviewer (YX) screened all identified titles and abstracts. Full texts of relevant documents were obtained, read, and assessed for relevance by this reviewer. The authors of a few potentially eligible studies were contacted by emails with subjects/titles and contents drafted in a similar manner, for their published full-text reports. Further literature was sought through the reference lists and citation trials of eligible studies. Each included study was identified on the Web of Science database, from where studies on the reference list and subsequent studies that cited it were exported to Endnote, followed by the same title, abstract, and full-text screening process.

## Data extraction and quality assessment

Data extraction was completed by one reviewer (YX) using pre-specified data collection forms, and all extractions were checked by a second reviewer (ZZ). Extracted data were country, year of publication, author, recruiting sites and periods, case selection, study design, sample size, definition for smoking, frequency of women and men who never smoked or were former or current smokers, definitions or diagnostic criteria used for PAD, and frequency of participants with PAD. When relevant information was not reported in a study, we presented it as missing value. When more than one multivariable adjustment was carried out, we extracted the one with the most covariates. The Newcastle-Ottowa Scale [7] was adopted for quality assessment. This tool contains eight internal validity items and three core domains, and has been assessed as one of the 6 "best" tools that can be used in a systematic review for cohort studies [8]. This scale has also been adapted to be used in cross-sectional studies [9]. For the domain of "comparability", we pre-specified age and socioeconomic status (SES) as the most important factors to be controlled for. The reasons for SES to be controlled for were: 1) SES is related to both smoking status and PAD risk [10,11]; and 2) the relationship between smoking and PAD is unlikely to be mediated through SES. Quality of included studies was independently assessed by two reviewers (YX, ZZ). Interrater reliability was measured by Cohen's kappa. Results were compared and discrepancies were solved by mutual consent.

## Statistical analysis

For each study, the sex-specific HRs, RRs, or ORs for PAD were obtained. HRs and RRs were considered as similar measures and thus were combined as RRs. For each study, sex-specific estimates of the association and 95% CIs were used to calculate the women-to-men comparisons (ratio of RRs or ORs, RRRs or RORs, and 95% CIs) [12]. Pooled estimates across studies were obtained using random-effects models. Studies were weighted according to the inverse of the variance of log RRs or log ORs, and log RRRs or log RORs. The $I^2$ statistic was used to estimate the percentage of variability among studies attributable to between-study heterogeneity, and the p-values for Cochran's Q test were also reported. Subgroup analysis, meta-regression, sensitivity analyses, and publication bias were not conducted or assessed, as the number of studies in each comparison was small. Meta-analyses were conducted in Stata/MP 18.0 and results were visualized using R 4.2.2.

## Results

After removing duplicates 4,672 records were identified, and 4,372 and 283 records were excluded at title and abstract, and full-text screening stages, respectively, (Fig 1, S2 File). Four studies that did not report estimates of the variance were excluded. Three of them were published in 1980s and one in 1994 with no valid contact information for the authors. Seventeen studies (all published in English) were included (S2A–S2E Table and S3 File). Fourteen studies were conducted in the World Bank defined high-income countries (Australia [13], Finland [14], Norway [15], Spain [16–19], UK [20–22], and USA [23–26]). Three studies were conducted in an upper-middle income country, China [27–29]. All extracted information (S2A–S2D Table and Figs 2 and 3) were found in the studies. There were some missing data in S2E Table, as the numbers of former, current, and/or never smokers among women, men, and/or the whole study sample were not reported in a few studies [14,17,21,24–26,28,29].

There were four cohort studies (published between 2015 and 2023, 2,117,860 participants, 54.4% women) [13,20–22], totaling 21,989 (1.0%) incident cases of PAD (women 9,112/1,152,613, 0.8%; men 12,877/965,247, 1.3%). All studies used hospital inpatient data and death registrations to identify PAD [13,20–22]. Primary care consultation data were an additional source in one study [20]. The lengths of follow-ups were reported as a mean of 7.2 [13] or 19.9 years [21], or a median of 6 [20] or 12.6 years (interquartile interval 11.8, 13.3 years) [22]. Hospitalized or fatal PAD were identified using International Classification of Diseases and/or procedure codes in all cohort studies (S2B Table for codes) [13,20–22]. One study used Health Data Research UK's PAD phenotyping definitions and coding system and additionally identified PAD diagnoses in primary care [20].

Thirteen studies were cross-sectional (published between 2000 and 2023, 230,436 participants, 59.9% women) [14–19,23–29]. Nine of these defined PAD as ABI of < or ≤0.9 [14,16–18,23,24,26–28] including one study which also classified those with intermittent claudication according to the WHO/Rose questionnaire (regardless of ABI) as PAD [27]. Two studies used ultrasound to identify stenosis, occlusion, or plaque [19,29]. One study used a Norwegian translation of the Edinburgh Claudication Questionnaire to identify participants with intermittent claudication [15]. Another study identified symptomatic PAD in a geriatric practice [25]. Frequency of cases was 6.9% (15,996/230,436) (women 9,352/137,962, 6.8%; men 6,644/92,474, 7.2%).

## Quality assessment

Cohen's kappa was 0.42 for 28 items with 2 levels and 1 for 4 items with 3 levels in cohort studies. It was 0.46 for 52 items with 2 levels and 0.52 for 39 items with 3 levels in cross-sectional

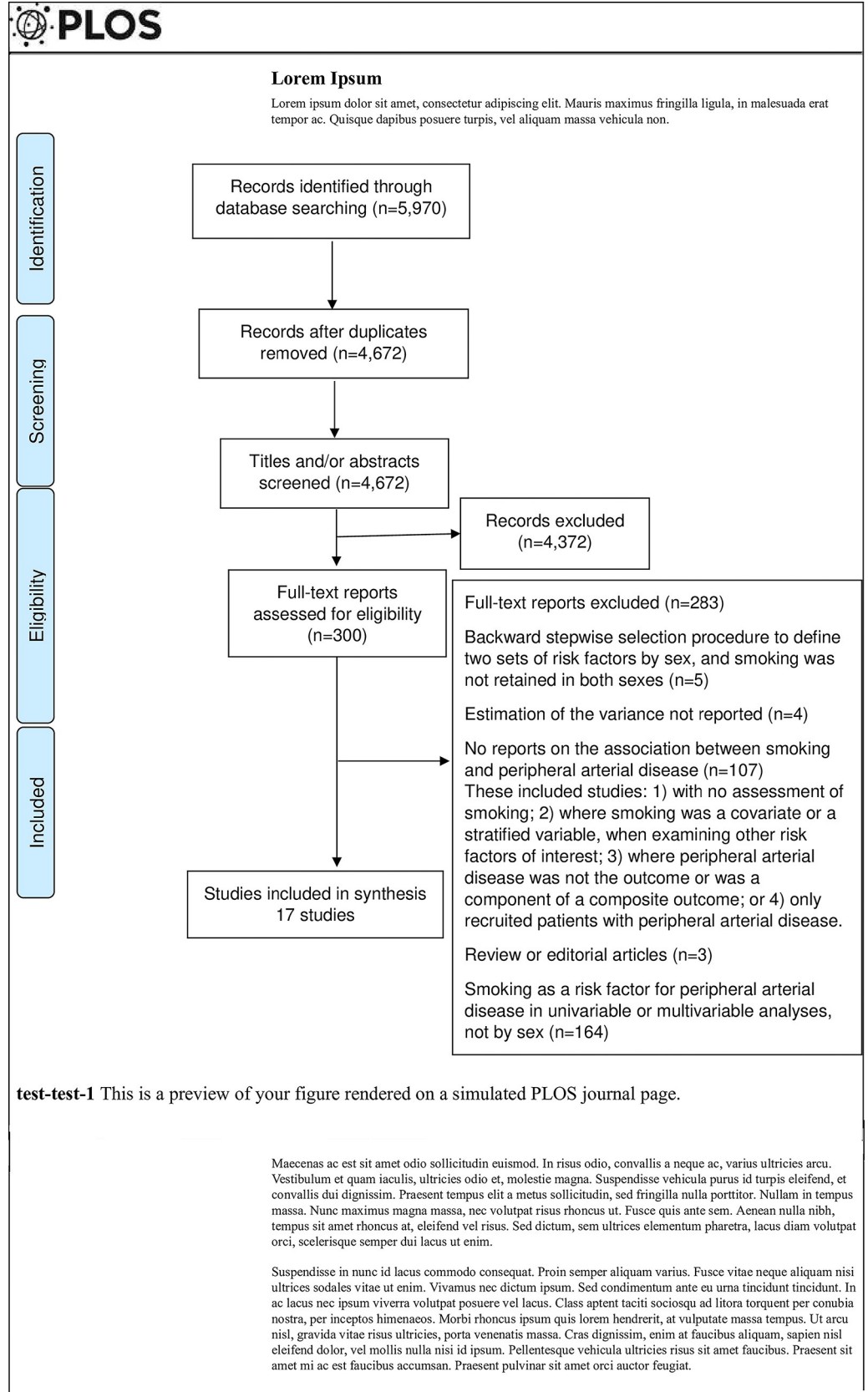

**Fig 1. Flow diagram for systematic review.**

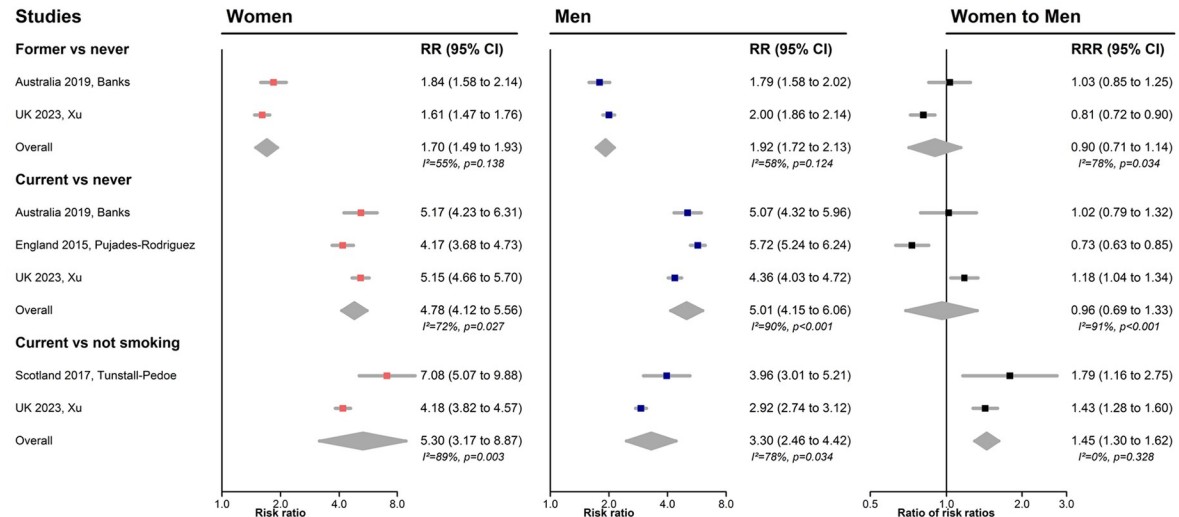

**Figure 2_202312200124.tif** This is a preview of your figure rendered on a simulated PLOS journal page.

Maecenas ac est sit amet odio sollicitudin euismod. In risus odio, convallis a neque ac, varius ultricies arcu. Vestibulum et quam iaculis, ultricies odio et, molestie magna. Suspendisse vehicula purus id turpis eleifend, et convallis dui dignissim. Praesent tempus elit a metus sollicitudin, sed fringilla nulla porttitor. Nullam in tempus massa. Nunc maximus magna massa, nec volutpat risus rhoncus ut. Fusce quis ante sem. Aenean nulla nibh, tempus sit amet rhoncus at, eleifend vel risus. Sed dictum, sem ultrices elementum pharetra, lacus diam volutpat orci, scelerisque semper dui lacus ut enim.

Suspendisse in nunc id lacus commodo consequat. Proin semper aliquam varius. Fusce vitae neque aliquam nisi ultrices sodales vitae ut enim. Vivamus nec dictum ipsum. Sed condimentum ante eu urna tincidunt tincidunt. In ac lacus nec ipsum viverra volutpat posuere vel lacus. Class aptent taciti sociosqu ad litora torquent per conubia nostra, per inceptos himenaeos. Morbi rhoncus ipsum quis lorem hendrerit, at vulputate massa tempus. Ut arcu nisl, gravida vitae risus ultricies, porta venenatis massa. Cras dignissim, enim at faucibus aliquam, sapien nisl eleifend dolor, vel mollis nulla nisi id ipsum. Pellentesque vehicula ultricies risus sit amet faucibus. Praesent sit amet mi ac est faucibus accumsan. Praesent pulvinar sit amet orci auctor feugiat.

Phasellus vitae congue est. Duis rutrum iaculis nunc, sed sollicitudin neque eleifend nec. Pellentesque ac nisi eget tortor imperdiet sagittis ut in orci. Mauris porta convallis euismod. Donec in ultricies urna, nec interdum lectus. Nullam sit amet finibus augue, eget rutrum metus. Nam faucibus, urna ac finibus eleifend, neque nisi lobortis ante, at pharetra purus purus sed urna. Curabitur sit amet dui at enim porta posuere non vehicula ligula. Suspendisse potenti. Vestibulum arcu magna, vulputate a massa ac, molestie tincidunt dui.

Donec id tempus lacus, sed tristique nulla. Nullam rutrum risus ut pharetra porttitor. Nam mattis dolor erat, sed volutpat est mattis sed. Suspendisse eu porta tellus. Cras gravida velit sed maximus fermentum. Fusce vitae metus commodo, sagittis nunc sed, faucibus nunc. Integer iaculis quam mattis, luctus neque in, viverra magna. Nulla rhoncus feugiat orci, quis posuere ligula ornare at. Integer vel sagittis risus. Donec semper metus nec finibus accumsan. Mauris sit amet suscipit ante. Aliquam accumsan, nisl vitae vulputate elementum, turpis nibh varius urna, vel bibendum nulla nunc ac quam. Aenean malesuada egestas maximus. Pellentesque faucibus, odio at tincidunt ullamcorper, eros nisi pellentesque mi, non blandit sapien neque quis lectus.

**Fig 2. Associations between smoking and peripheral artery disease and sex comparison of associations in cohort studies.** CI confidence interval, RR risk ratio, RRR ratio of risk ratios. Adjusted variables: Australia 2019, Banks age, region of residence, alcohol consumption, household income, and education; England 2015, Pujades-Rodriguez age; UK 2023, Xu age, socioeconomic status, body mass index; Scotland 2017, Tunstall-Pedoe age. There are nine separate meta-analyses. In each meta-analysis, weight of each study is calculated based on the inverse of within study variances. That is, a study with a narrower confidence interval was weighted greater than a study with a wider confidence interval.

studies. After discussion, two reviewers reached agreements for all ratings in the quality assessments. Four cohort studies were of good quality, all scoring 8 of a possible maximum 9 points (S3A Table). Five cross-sectional studies were of good quality, scoring 6 or 7 of a possible maximum 9 points (S3B Table). The other eight cross-sectional studies [14,15,18,19,24,26,28,29] scored 2 to 5, due to their samples being non-representative of the general population, unjustified sample size, lack of comparisons between respondents and non-respondents, assessment of exposure (smoking status, etc.) not detailed, not controlling for SES, and unblinded assessment of outcomes.

## Associations reported in cohort studies

One study reported RRs [13], whereas HRs were reported in the other three studies [20–22]. Compared to those who never smoked, former [13,22] or current smokers [13,20,22] had a higher risk of PAD (Fig 2), similarly in women and men. Combining data from two studies of 15,737 [21] and 500,207 participants [22], we found compared to those who never or previously smoked, current smokers had a higher risk of PAD (combined RR 5.30 (3.17, 8.87) in women and 3.30 (2.46, 4.42) in men) [21,22]. The excess risk was higher in women than men: the combined RRR was 1.45 (1.30, 1.62), with no evidence of heterogeneity between the estimates ($I^2 = 0\%$, $p = 0.328$). One study also compared former with current smokers, where the women-to-men ratio of HRs was 0.69 (0.61, 0.78) [22]. One study examined the relationship between smoking intensity or toxic chemicals and the risk of PAD [21]. Per one additional cigs/day and per one ppm higher expired carbon monoxide were respectively related to 20% and 17% higher excess risk of PAD, in women compared to men (S4 Table).

## Associations reported in cross-sectional studies

The comparisons were between current (or former) and never smokers in seven studies (Fig 3) [15,17–19,23,27,29], and between current and not smoking (those who quit or who never smoked) in seven studies [14,16,23–26,28]. These included one study that compared former and never smokers, as well as current and not smoking groups [23]. Compared to those who never smoked, women former and current smokers had a lower excess risk than men counterparts: combined women-to-men ROR 0.64 (0.46 to 0.90) and 0.63 (0.50, 0.79), respectively. There was no heterogeneity between the estimates ($I^2 = 30\%$, $p = 0.192$, and $I^2 = 0\%$, $p = 0.594$, respectively). Current smoking (compared to never smoked or to not smoking) was related to a higher risk of PAD, with no evidence of a sex difference.

Three studies examined the relationship between smoking intensity and/or duration (pack-years, cigarettes per day, years of smoking), years since smoking cessation, or age of starting smoking and the risk of PAD (S4 Table) [15,19,27]. For example, one study reported that participants who had quit smoking for over 20 years had lower prevalence of intermittent claudication than current smokers, with no sex difference ORs 0.4 (0.2, 0.8) in women and 0.2 (0.1, 0.5) in men [15]. Quantitative synthesis was not possible due to the inconsistency in measurements. No sex difference in the relationships were found.

## Lorem Ipsum

Lorem ipsum dolor sit amet, consectetur adipiscing elit. Mauris maximus fringilla ligula, in malesuada erat tempor ac. Quisque dapibus posuere turpis, vel aliquam massa vehicula non.

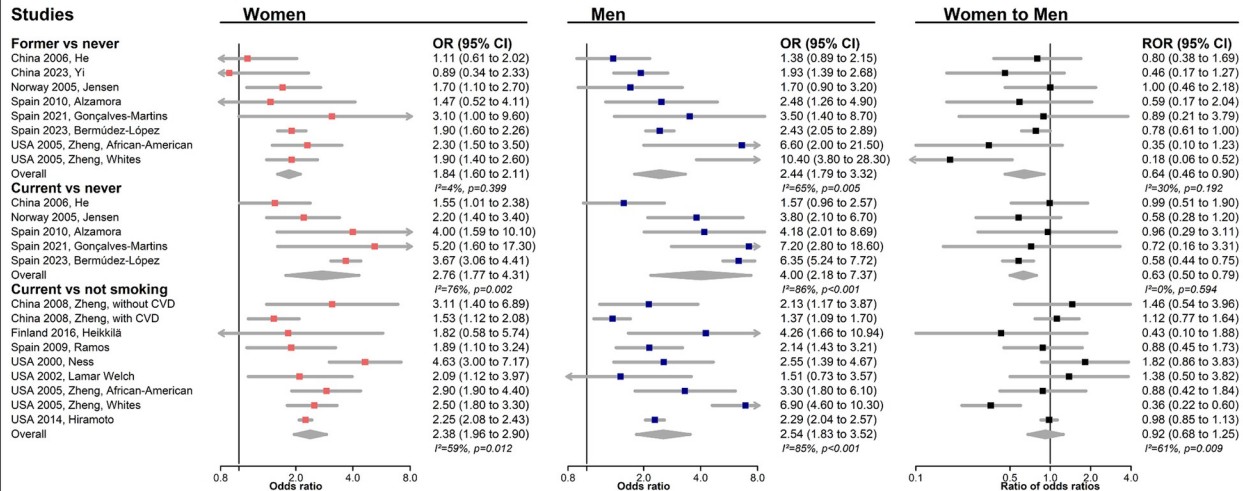

**test-test-1** This is a preview of your figure rendered on a simulated PLOS journal page.

Maecenas ac est sit amet odio sollicitudin euismod. In risus odio, convallis a neque ac, varius ultricies arcu. Vestibulum et quam iaculis, ultricies odio et, molestie magna. Suspendisse vehicula purus id turpis eleifend, et convallis dui dignissim. Praesent tempus elit a metus sollicitudin, sed fringilla nulla porttitor. Nullam in tempus massa. Nunc maximus magna massa, nec volutpat risus rhoncus ut. Fusce quis ante sem. Aenean nulla nibh, tempus sit amet rhoncus at, eleifend vel risus. Sed dictum, sem ultrices elementum pharetra, lacus diam volutpat orci, scelerisque semper dui lacus ut enim.

Suspendisse in nunc id lacus commodo consequat. Proin semper aliquam varius. Fusce vitae neque aliquam nisi ultrices sodales vitae ut enim. Vivamus nec dictum ipsum. Sed condimentum ante eu urna tincidunt tincidunt. In ac lacus nec ipsum viverra volutpat posuere vel lacus. Class aptent taciti sociosqu ad litora torquent per conubia nostra, per inceptos himenaeos. Morbi rhoncus ipsum quis lorem hendrerit, at vulputate massa tempus. Ut arcu nisl, gravida vitae risus ultricies, porta venenatis massa. Cras dignissim, enim at faucibus aliquam, sapien nisl eleifend dolor, vel mollis nulla nisi id ipsum. Pellentesque vehicula ultricies risus sit amet faucibus. Praesent sit amet mi ac est faucibus accumsan. Praesent pulvinar sit amet orci auctor feugiat.

**Fig 3. Cross-sectional associations between smoking and peripheral artery disease and sex comparison of associations.** CVD denotes cardiovascular disease, CI confidence interval, OR odds ratio, ROR ratio of odds ratios. Adjusted variables: China 2006 He age, marital status, education (≤6, 7–12, ≥13 years), alcohol drinking (current drinkers or not), exercise (<1, 1–3, ≥4 hours/day), body mass index (BMI), hypertension or diabetes, and family histories of coronary heart disease (CHD) or stroke; China 2023 Yi age, diabetes, low density lipoprotein cholesterol (LDL-C), lipoprotein(a); Finland 2016, Heikkilä age, height, waist circumference, pulse pressure, fasting glucose, total cholesterol (TC); Norway 2005 Jensen age; Spain 2010, Alzamora age, physical activity (no limitation, mild limitation, only able light activity, or breathless any activity), >7h walking per week, BMI, hypertension, hypercholesterolemia, high triglycerides, diabetes, CVD, and recruitment center; Spain 2021, Gonçalves-Martins high blood pressure for women, diabetes for men; China 2008, Zheng age, TC, LDL-C, fasting glucose, uric acid, obesity; Spain 2009, Ramos age, CVD, diabetes, and uncontrolled hypertension for women; age, CVD, definite or atypical claudication (Edinburgh questionnaire), and uncontrolled hypertension for men; Spain 2023, Bermúdez-López age, hypertension, obesity, dyslipidemia, prediabetes or diabetes, Mediterranean diet adherence score, neck perimeter and abdominal obesity; USA 2000, Ness age, hypertension, diabetes, high-density lipoprotein cholesterol (HDL-C) and LDL-C; USA 2002 Welch age; USA 2005, Zheng age, LDL-C, hypertension, and diabetes; USA 2014, Hiramoto age, race, hypertension, smoking, C-reactive protein, CHD, TC/HDL-C ratio, and diabetes. There are nine separate meta-analyses. In each meta-analysis, weight of each study is calculated based on the inverse of within study variances. That is, a study with a narrower confidence interval was weighted greater than a study with a wider confidence interval.

## Discussion

Smoking is a well-known independent risk factor for PAD, yet few studies have quantified sex-specific relationships between smoking and PAD. In our pooled analyses, we showed that both lifetime abstinence and quitting smoking were associated with a lower risk of PAD, broadly in much the same way in both women and men. In cohort analyses, there was some evidence of a greater PAD risk, in women than in men, from continuing to smoke, compared to never having smoked or having quit. However, compared to never smoking, based on limited evidence from cross-sectional studies, former and current smoking (compared to those never smoked) were related to a lower excess risk in women than in men.

The most recent meta-analyses [2] compared the risk of PAD between 1) current smokers and those who were not currently smoking, and 2) former and never smokers. For the comparison between current smoking and not smoking, our combined OR from cross-sectional studies of 2.38 (1.96, 2.90) in women and 2.54 (1.83, 3.52) in men were similar to the 2.71 (2.28, 3.21) reported previously [2]. Yet, the previous estimate [2] combined women and men, and cohort and cross-sectional evidence. Our synthesis of cohort studies obtained greater combined estimate/RR, 5.30 (3.17, 8.87) in women and 3.30 (2.46, 4.42) in men, which could be due to two reasons. First, the increased benefits of smoking cessation as the time since quitting increased [30]. Second, new evidence from cohort studies (all published after the previous evidence synthesis) was added in the current work. This includes a major difference in the methodology used to ascertain PAD in the cohort studies: hospitalized or fatal PAD [13,20–22], or PAD diagnoses in primary care [20] in the current work, versus PAD defined using ABI and/or questionnaires in previous cohort studies [2]. For the comparison between former and never smokers, our estimates for women and men in cohort studies and for women in cross-sectional studies were close to the combined OR 1.67 (1.54, 1.81) [2]. We found that the increased risk of PAD in men former smokers compared to never smokers in cross-sectional studies was greater, OR 2.44 (1.79, 3.32). However, reverse causality is likely. That is, men with PAD might be more likely to quit than women with PAD, due to greater concerns of own health than women [31].

Some under- or over-estimations should be noted. Among smokers who were in their 60s, both recent smoking habits and those in early adult life over 40 years ago, have been related to mortality [32]. Thus, to measure excess hazards for women smokers in countries such as the UK or USA, where smoking prevalence in young women did not peak until the 1960s (decades later than in men), it requires follow-up to be more than 40 years later/after 2000 [32]. Otherwise, full eventual risks of smoking might be underestimated [32]. Accordingly, in one of the included cohort studies, the follow-up might have been completed before 2000, especially for

some participants recruited before 1987 [21]. Yet, since the comparison in this study was between current and former or never smokers, any potential underestimation of risk in women, should have occurred among both current smokers (exposure group) and former smokers (part of the comparison group).

Further, smoking status might change during follow-up, but none of the cohort studies took this into account. In only four cross-sectional studies [16,18,19,27], former smokers were defined as those who stopped smoking for at least 30 days [19], or one [16,18] or two years [27], whereas in the remaining studies, it was unclear whether former smokers had achieved long-term abstinence. This is important, as there is evidence to suggest sex difference in quitting attempts and success. Although women were 25% more likely than men to make a quit attempt [33], they were less likely than men to maintain smoking abstinence at one year after a quit attempt [34]. Consequently, in cohort studies, estimations for the comparisons between former and never smokers might have been overestimated, especially in women, leading to overestimation of women-to-men RRR. Thus, if the combined women-to-men RRR of 0.90 (0.71, 1.14) was an overestimation, women who quit might have a lower excess risk of PAD than men counterparts. Conversely, estimates for the comparisons between current smokers and not smokers (defined as never and/or former smokers) might have been underestimated to a greater degree in women than in men, resulting in underestimation of women-to-men RRR.

Overall, sex differences were neglectable, at least inconsistent, with women and men being equally affected by smoking in half of the comparisons. CHD, stroke (at least ischemic stroke), and PAD are considered as similar atherosclerotic diseases affecting different vascular territories. Although a stronger deleterious effect of smoking in women than men smokers was found for CHD [3], this was not true for stroke where equal hazardous effects were found in women and men [35]. A possible explanation given in the stroke study was the antiestrogenic effect in women smokers, adversely affecting lipid profile, a major risk factor for CHD but to a lesser extent for stroke [35]. This may also apply to PAD, for which the importance of lipids was replaced by inflammation [21]. That said, there is still an indication of a sex difference in the risk of stroke related to smoking, as in Western populations, women smokers were found to have a 10% greater excess risk of stroke compared to men smokers [35].

Finding no sex difference is remarkable, for two reasons. The first is that women are known to have had a shorter duration of smoking, and at a lower intensity. Women in the current systematic review were likely to have begun smoking at an older age than did men. This is evidenced by the Global Adult Tobacco Survey conducted between 2008 and 2010, where women aged 45 years and over at the time of the survey began smoking at an older age than did equivalently aged men [36]. Pooling data from 13 countries, it was estimated that women daily smokers smoked fewer cigarettes per day than their men counterparts (mean difference: -3.78 (-4.71, -2.85)) [36,37]. It was also reported that women take smaller puffs of shorter duration and leave longer butts compared with men [38]. Second, our findings of no sex difference might have been based on the possibility of sex differentiated risk in the women and men never smokers and/or those who quit (the comparison groups). For example, in analyses of never smokers and those who quit for over 10 years [39] or of never smokers alone [23], women were much more likely than men to have ABI of <1.0 or of ≤0.9. Thus, same amount of exposure to smoking might have taken more harmful effects in women than in men, for some comparisons of no sex difference to be observed.

PAD is related to more extreme consequences in women than in men. For instance, women with PAD compared with men counterparts had poorer initial functional performance and greater functional decline [40]. When presenting for lower limb revascularization, women were more likely to have more severe symptoms of chronic limb-threatening ischemia rather

than the mild to moderate symptoms of claudication [41]. After lower limb revascularization, women had inferior 30-day outcomes (higher rates of mortality, amputation, early graft thrombosis, embolization, cardiac events, and stroke) compared with men [42]. Among people who had nontraumatic transtibial or transfemoral amputation, women were 40% more likely to receive transfemoral amputation than men [43]. These adverse consequences in women are likely due to women being asymptomatic or having atypical symptoms at early stages [44], and greater missed diagnoses in women. Thus, the importance of risk factor identification and modification in primary practice should be emphasized, prior to and post a PAD diagnosis. Our results suggested at least the same value of smoking abstinence in women and men, considering the observed equal hazardous effects or inconsistencies in the direction of sex differences.

## Strengths and limitations

To the best of our knowledge, this is the first systematic review to synthesize evidence on smoking as a risk factor for PAD that compares women and men. There are some limitations in this systematic review. First, the literature search, screening, and data extraction were conducted by one reviewer, although reference lists and citation trails were sourced to capture studies that may be missed out, and data extraction was checked multiple times by this reviewer and by a second reviewer. Second, Cohen's kappa between two reviewers for quality assessment was largely in the 0.4 to 0.59 range, indicating moderate level of agreement [45]. Third, we did not include studies on passive cigarette, pipes, cigars smoking, nor active or passive smoking of e-cigarettes/vapes, although there were no studies that were excluded due to this reason. Additionally, limitations in prior studies included some selection and measurement bias. Given that smokers could have died from smoking at younger ages before entering to the studies, the participants in our systematic review were survivors. Second, study results were generally heterogeneous, possibly due to variability in the definitions for smoking and PAD, case mix, percentage of women who smoked, recruitment year, follow-up time, and variables accounted for, etc. Finally, due to the paucity of studies reporting the risk by sex, we cannot examine how the mentioned source of heterogeneities impact on the women-to-men difference.

## Conclusions

A higher risk of PAD in former or current smokers than in those who never smoked was found in both cohort and cross-sectional studies. Evidence from longitudinal studies suggested that current smoking (compared to not smoking) may be related to greater hazardous effects on developing PAD in women compared to men. That is, continuing to smoke may result in a greater excess risk of PAD, whereas smoking cessation, a greater reduced risk, in women than men. The key message remains that women and men should equally be discouraged to start smoking and encouraged to quit if they have already smoked.

## Supporting information

**S1 Checklist.**
(DOCX)

**S1 Table. Description of search strategy and results (3 March 2024, n = 4,672 after removing duplicates).**
(PDF)

**S2 Table. a. Characteristics of included studies (recruitment, study population and design, inclusion and exclusion criteria, and age of participants).** AAA denotes abdominal aorta aneurysm, ABI ankle brachial index, CRP C-reactive protein, CVD cardiovascular disease, IC intermittent claudication, NSW New South Wales, PAD peripheral artery disease, SD standard deviation, USA United States of America a. Study base: C community-based, H hospital-based, P population-based b. Study design: C cohort, X cross-sectional. Please refer to S3 File for the refences of studies. **b. Characteristics of included studies (smoking and peripheral artery disease identification).** ABI denotes ankle brachial index, CVD cardiovascular disease, IC intermittent claudication, ICD International Classification of Diseases, ICD-10-AM International Statistical Classification of Diseases and Related Health Problems, Tenth Revision, Australian Modification, OPCS 4 Classification of Interventions and Procedures, PAD peripheral artery disease, SBP systolic blood pressure, WHO World Health Organization. Please refer to S3 File for the refences of studies. **c. Characteristics of included studies (sample sizes).** PAD denotes peripheral artery disease. Please refer to S3 File for the refences of studies. **d. Characteristics of included studies (sample sizes by PAD status and sex).** PAD denotes peripheral artery disease. Please refer to S3 File for the refences of studies. **e. Characteristics of included studies (sample sizes by smoking status and sex).** PAD denotes peripheral artery disease. Please refer to S3 File for the refences of studies.
(PDF)

**S3 Table. a. Quality assessment of cohort studies using the Newcastle-Ottawa scaleTable e-3b Quality assessment of cross-sectional studies using the Newcastle-Ottawa scale.** Please refer to S3 File for the refences of studies. **b. Table Associations between other measures of smoking and the risk of peripheral artery disease.** BMI denotes body mass index, CHD coronary heart disease, HDL-C high-density lipoprotein cholesterol, HR hazard ratio, OR odds ratio, RHR ratio of hazard ratio, ROR ratio of odds ratio, SBP systolic blood pressure, SES socioeconomic status, TC total cholesterol, T1 the lowest tertial, T2 the second tertial, T3 the highest tertial. [1]cut-off for tertials of duration of smoking were 18 years for women and 30 years for men. [2]cut-off for tertials of pack-years were 10 and 16 pack-years for women, and 13 and 22 for men. [3]cut-off for tertials of pack years were 3.0 and 8.3 pack-years for women, and 6.6 and 15.0 for men. [4]cut-off points for tertials of time since quitting were 10 and 20 years for women, and 9 and 20 for men. [5]cut-off points for tertials of age when started smoking were 18 and 20 years for women, and 16 and 19 for men. Please refer to S3 File for the refences of studies.
(PDF)

**S4 Table. Associations between other measures of smoking and the risk of peripheral artery disease.**
(PDF)

**S1 File.** a. Preferred Reporting Items for Systematic Reviews and Meta-Analyses (PRISMA) 2020 for abstracts checklist. b. Preferred Reporting Items for Systematic Reviews and Meta-Analyses (PRISMA) 2020 Checklist. c. Meta-analysis Of Observational Studies in Epidemiology (MOOSE) Guideline.
(PDF)

**S2 File. List of excluded studies in full-text screening stage with reasons (n = 283).**
(PDF)

**S3 File. List of included studies (n = 17).**
(PDF)

## Author Contributions

**Conceptualization:** Mark Woodward.

**Formal analysis:** Ying Xu.

**Funding acquisition:** Ying Xu, Mark Woodward, Katie Harris.

**Investigation:** Ying Xu.

**Methodology:** Ying Xu, Anna Louise Pouncey, Zien Zhou, Mark Woodward, Katie Harris.

**Supervision:** Mark Woodward, Katie Harris.

**Validation:** Ying Xu, Zien Zhou.

**Visualization:** Ying Xu, Katie Harris.

**Writing – original draft:** Ying Xu.

**Writing – review & editing:** Ying Xu, Anna Louise Pouncey, Zien Zhou, Mark Woodward, Katie Harris.

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
