## [Decision Letter · Decision Letter 0]

15 Feb 2024

PONE-D-23-43121Smoking as a risk factor
for lower extremity peripheral artery disease in women compared to men: a systematic
review and meta-analysisPLOS ONE

Dear Dr. Xu,

Thank you for submitting your manuscript to PLOS ONE. After careful consideration, we
feel that it has merit but does not fully meet PLOS ONE’s publication criteria as it
currently stands. Therefore, we invite you to submit a revised version of the
manuscript that addresses the points raised during the review process.

Please submit your revised manuscript by Mar 31 2024 11:59PM. If you will need more
time than this to complete your revisions, please reply to this message or contact
the journal office at plosone@plos.org. When
you're ready to submit your revision, log on to https://www.editorialmanager.com/pone/ and select the 'Submissions
Needing Revision' folder to locate your manuscript file.

Please include the following items when submitting your revised
manuscript:A rebuttal letter that responds to each point raised by the academic
editor and reviewer(s). You should upload this letter as a separate file
labeled 'Response to Reviewers'.A marked-up copy of your manuscript that highlights changes made to the
original version. You should upload this as a separate file labeled
'Revised Manuscript with Track Changes'.An unmarked version of your revised paper without tracked changes. You
should upload this as a separate file labeled 'Manuscript'.If you would like to make changes to your financial disclosure,
please include your updated statement in your cover letter. Guidelines for
resubmitting your figure files are available below the reviewer comments at the end
of this letter.

We look forward to receiving your revised manuscript.

Kind regards,

Athanasios Saratzis

Academic Editor

PLOS ONE

“MW has done recent consultancy for Amgen and Freeline outside the submitted work, no
support from any organization, for the submitted work, no other relationships or
activities that could appear to have influenced the submitted work. All other
authors have nothing to declare.”

4. In the online submission form you indicate that your data is not available for
proprietary reasons and have provided a contact point for accessing this data.
Please note that your current contact point is a co-author on this manuscript.
According to our Data Policy, the contact point must not be an author on the
manuscript and must be an institutional contact, ideally not an individual. Please
revise your data statement to a non-author institutional point of contact, such as a
data access or ethics committee, and send this to us via return email. Please also
include contact information for the third party organization, and please include the
full citation of where the data can be found.

Reviewers' comments:

Reviewer's Responses to Questions

**Comments to the Author**

1. Is the manuscript technically sound, and do the data support the conclusions?

Reviewer #1: Yes

Reviewer #2: Yes

2. Has the statistical analysis been performed
appropriately and rigorously? 

Reviewer #1: Yes

Reviewer #2: Yes

3. Have the authors made all data underlying the
findings in their manuscript fully available?

Reviewer #1: Yes

Reviewer #2: Yes

4. Is the manuscript presented in an intelligible
fashion and written in standard English?

Reviewer #1: Yes

Reviewer #2: Yes

5. Review Comments to the Author

Reviewer #1: Thanks for the opportunity to review this manuscript reporting the
results of a meta-analysis of studies reporting whether the relationship between
smoking and peripheral arterial disease differs by sex. The authors have carried out
a rigorous review of available literature and a statistically sound meta-analysis.
Prior to publication, I recommend a few minor edits to be made:

Introduction/Discussion:

• Important to explain to the reader that there is significant under-recognition of
cardiovascular risk in women and failure to recognise the importance of secondary
prevention, sex-related differences in clinical presentation, and delay in
presentation and misperception of cardiovascular disease in women. These factors
contribute to peripheral arterial disease being underdiagnosed and understudied in
women (see doi:10.1007/S11883-018-0742-X and PMID: 34010613). As well as in the
introduction, it is worth emphasising this point in the discussion: Lines 313 –
326.

Methodology:

• Line 82 - Authors mention passive smoking is not included yet they have included it
in their search strategy

• Increased risk of bias given that only one reviewer has screened and identified
articles – please discuss this in the limitations of the manuscript. (see: https://doi.org/10.1016/j.jclinepi.2020.01.005)

• The authors state that they contacted individuals of potentially eligible studies.
Please include how this was carried out and whether it was standardised.

Results:

• Line 149 & 150 – Please include SD of mean follow up & IQR of median follow
up of the studies.

Reviewer #2: The authors submitted a systematic review and meta-analysis concerning
smoking as risk factor for PAD in women related to men. The methodology is sound,
and the conclusions justified. The discussion is weighty but addresses a variety of
sex related differences in PAD. I think this paper excellent and is an important
addition to the literature, complements on their work.

Thanks to authors for adhering to their Prospero submission. One major comment:

Line 100: The authors report only one reviewer screening all the documents by title,
abstract and full text. This one reviewer also performed all the extractions with a
second reviewer. Please account for this in your limitation section, as it
introduces bias.

Minor comments:

Line 86: diagnostic and procedure codes, are these listed anywhere, can they be
included?

Line 103: did you use a programme for citation review?

Line 110: how much missing data did you encounter, is this reported in the
results?

Line 120: did you look at the level of agreement between your two reviewers? i.e. via
cohen kappa.

Line 143: may be worth stating this an upper-middle income country still.

Line 326: this sentence can strengthened with the result included within the
sentence.

6. PLOS authors have the option to publish the peer
review history of their article (what does this mean?). If published, this will
include your full peer review and any attached files.

If you choose “no”, your identity will remain anonymous but your review may still be
made public.

**Do you want your identity to be public for this peer review?** For
information about this choice, including consent withdrawal, please see our
Privacy Policy.

Reviewer #1: **Yes: **Sarah Jane Messeder

Reviewer #2: No

---

## [Author Response · Author response to Decision Letter 0]

6 Mar 2024

Reviewer #1: Thanks for the opportunity to review this manuscript reporting the
results of a meta-analysis of studies reporting whether the relationship between
smoking and peripheral arterial disease differs by sex. The authors have carried out
a rigorous review of available literature and a statistically sound meta-analysis.
Prior to publication, I recommend a few minor edits to be made:

Introduction/Discussion:

• Important to explain to the reader that there is significant under-recognition of
cardiovascular risk in women and failure to recognise the importance of secondary
prevention, sex-related differences in clinical presentation, and delay in
presentation and misperception of cardiovascular disease in women. These factors
contribute to peripheral arterial disease being underdiagnosed and understudied in
women (see doi:10.1007/S11883-018-0742-X and PMID: 34010613). As well as in the
introduction, it is worth emphasising this point in the discussion: Lines 313 –
326.

Response: Thank you for this good point. Other than the paragraph in Discussion, we
have added more background for PAD in women in the Introduction, and now it reads
“The importance of cardiovascular risk in women is under-recognized, especially for
peripheral artery disease (PAD). Due to higher rates of asymptomatic disease or
atypical symptoms in women with PAD, they are often diagnosed at later stages of the
disease, and thus are less likely to receive interventions to prevent advanced PAD
and adverse outcomes of amputation, coronary heart disease (CHD), and stroke [1].”
(lines 50 to 54)

Methodology:

• Line 82 - Authors mention passive smoking is not included yet they have included it
in their search strategy

Response: This is a good point too. We did include “Passive Smoking” as a subject
heading for one database CINAHL by mistake and it would unnecessarily increase the
number of identified records and workload. We updated our search on 3 March 2024 and
corrected this error.

In this search, we also removed “Limit S4 to Research or Journal Article, Human” in
the CINAHL, because applying limits to “Research or Journal Article” did not reduce
the number of identified records and the function of limit to “Human” was not
available in this search.

No additional studies were added into our review from this updated search, but there
were three additional records (listed below) that were excluded in the full-text
screening process (S2 File), under “No reports on the association between smoking
and peripheral arterial disease”:

Baretella O, Buser L, Andres C, Häberli D, Lenz A, Döring Y, et al. Association of
sex and cardiovascular risk factors with atherosclerosis distribution pattern in
lower extremity peripheral artery disease. Front Cardiovasc Med. 2023;10:1004003.
doi: 10.3389/fcvm.2023.1004003.

Martelli E, Zamboni M, Sotgiu G, Saderi L, Federici M, Sangiorgi GM, et al.
Sex-Related Differences and Factors Associated with Peri-Procedural and 1 Year
Mortality in Chronic Limb-Threatening Ischemia Patients from the CLIMATE Italian
Registry. J Pers Med. 2023;13(2). doi: 10.3390/jpm13020316.

Onofrei V, Adam CA, Marcu DTM, Leon MM, Cumpăt C, Mitu F, et al. Gender Differences
and Amputation Risk in Peripheral Artery Disease-A Single-Center Experience.
Diagnostics (Basel). 2023;13(19). doi: 10.3390/diagnostics13193145.

• Increased risk of bias given that only one reviewer has screened and identified
articles – please discuss this in the limitations of the manuscript. (see: https://doi.org/10.1016/j.jclinepi.2020.01.005)

Response: We have now added this as a limitation (lines 348 to 351) “First, the
literature search, screening, and data extraction were conducted by one reviewer,
although reference lists and citation trails were sourced to capture studies that
may be missed out, and data extraction was checked multiple times by this reviewer
and by a second reviewer.”

• The authors state that they contacted individuals of potentially eligible studies.
Please include how this was carried out and whether it was standardised.

“The authors of a few potentially eligible studies were contacted for their published
full-text reports.”

Response: We added more details on how and whether standardised (lines 104 to 106).
“The authors of a few potentially eligible studies were contacted by emails with
subjects/titles and contents drafted in a similar manner, for their published
full-text reports.”

As an example, here is an email to Professor Päivi Korhonen, email address:
Arto.heikkila@fi mnet.fi

Dear Professor Päivi Korhonen,

Re Heikkilä A, Venermo M, Kautiainen H, Aarnio P, Korhonen P. Short stature in men is
associated with subclinical peripheral arterial disease. Vasa. 2016
Nov;45(6):486-490. doi: 10.1024/0301-1526/a000566.

Please I may introduce myself. My name is Ying Xu. I am a Research Fellow at the
George Institute for Global Health and Conjoint Lecture at the University of New
South Wales in Sydney.

I am conducting a systematic review and meta-analysis on sex difference in smoking as
a risk factor for peripheral artery disease.

I find the above-mentioned article of yours probably meet our inclusion criteria, but
I cannot find the full text. May I please ask whether you would be happy to share
the published full-text PDF or a word accepted version?

Many thanks,

Ying

Results:

• Line 149 & 150 – Please include SD of mean follow up & IQR of median follow
up of the studies.

Response: we have now added the interquartile interval “(interquartile interval 11.8,
13.3 years)” (lines 160 to 161) for one study, but the numbers were not reported by
other studies.

Reviewer #2: The authors submitted a systematic review and meta-analysis concerning
smoking as risk factor for PAD in women related to men. The methodology is sound,
and the conclusions justified. The discussion is weighty but addresses a variety of
sex related differences in PAD. I think this paper excellent and is an important
addition to the literature, complements on their work.

Thanks to authors for adhering to their Prospero submission. One major comment:

Line 100: The authors report only one reviewer screening all the documents by title,
abstract and full text. This one reviewer also performed all the extractions with a
second reviewer. Please account for this in your limitation section, as it
introduces bias.

Minor comments:

Line 86: diagnostic and procedure codes, are these listed anywhere, can they be
included?

Response: This is related to cohort studies, and we have now added the following text
and point readers to S2b Table for codes “Hospitalized or fatal PAD were identified
using International Classification of Diseases and/or procedure codes in all cohort
studies (S2b Table for codes) [14, 21-23]. One study used Health Data Research UK’s
PAD phenotyping definitions and coding system and additionally identified PAD
diagnoses in primary care [21].” (lines 161 to 164) We also added “(search for
“PH236” on https://phenotypes.healthdatagateway.org/ )” in S2b Table for one
study, England 2015, Pujades-Rodriguez, as a completed list would be lengthy.

Line 103: did you use a programme for citation review?

Response: We did not use a separate programme, although we have added more details on
our citation review, which now it reads “Each included study was identified on the
Web of Science database, from where studies on the reference list and subsequent
studies that cited it were exported to Endnote, followed by the same title,
abstract, and full-text screening process.” (lines 107 to 110)

Line 110: how much missing data did you encounter, is this reported in the
results?

Response: In the Methods we now report that “When relevant information was not
reported in a study, we presented it as missing value.”, and in the Results that
”All extracted information (S2a-d Table, Figs 2 and 3) were found in the studies.
There were some missing data in S2e Table, as the numbers of former, current, and/or
never smokers among women, men, and/or the whole study sample were not reported in a
few studies [15, 18, 22, 25-27, 29, 30].” (lines 150 to 153)

Line 120: did you look at the level of agreement between your two reviewers? i.e. via
cohen kappa.

Response: We have now added Cohen’s kappa along with some interpretations for
reliability. In the Method, it reads “Interrater reliability was measured by Cohen’s
kappa.” (line 127)

In the Results, it reads “Cohen’s kappa was 0.42 for 28 items with 2 levels and 1 for
4 items with 3 levels in cohort studies. It was 0.46 for 52 items with 2 levels and
0.52 for 39 items with 3 levels in cross-sectional studies. After discussion, two
reviewers reached agreements for all ratings in the quality assessments.” (lines 175
to 178)

In the Discussion, “Second, Cohen’s kappa between two reviewers for quality
assessment was largely in the 0.4 to 0.59 range, indicating moderate level of
agreement [46]. (lines 351 to 353)

Line 143: may be worth stating this an upper-middle income country still.

Response: We made the sentence clearer as suggested, and it now reads “Three studies
were conducted in an upper-middle income country, China [27-29].” (line 149-150)

Line 326: this sentence can strengthened with the result included within the
sentence.

Response: We added the key finding to support this statement. Now, it reads “Our
results suggested at least the same value of smoking abstinence in women and men,
considering the observed equal hazardous effects or inconsistencies in the direction
of sex differences.” (lines 342 to 344)

to Reviewers.docx

---

## [Editor Report · Decision Letter 1]

8 Mar 2024

Smoking as a risk factor for lower extremity peripheral artery disease in women
compared to men: a systematic review and meta-analysis

PONE-D-23-43121R1

Dear Dr. Xu,

We’re pleased to inform you that your manuscript has been judged scientifically
suitable for publication and will be formally accepted for publication once it meets
all outstanding technical requirements.

Kind regards,

Athanasios Saratzis

Academic Editor

PLOS ONE
---

## [Editor Report · Acceptance letter]

1 Apr 2024

PONE-D-23-43121R1 

PLOS ONE

Dear Dr. Xu, 

I'm pleased to inform you that your manuscript has been deemed suitable for
publication in PLOS ONE. Congratulations! Your manuscript is now being handed over
to our production team.

Kind regards, 

on behalf of

Dr. Athanasios Saratzis 

Academic Editor

PLOS ONE